# Prenatal Cases Reflect the Complexity of the *COL1A1/2* Associated Osteogenesis Imperfecta

**DOI:** 10.3390/genes13091578

**Published:** 2022-09-02

**Authors:** Kai Yang, Yan Liu, Jue Wu, Jing Zhang, Hua-ying Hu, You-sheng Yan, Wen-qi Chen, Shu-fa Yang, Li-juan Sun, Yong-qing Sun, Qing-qing Wu, Cheng-hong Yin

**Affiliations:** 1Prenatal Diagnosis Center, Beijing Obstetrics and Gynecology Hospital, Capital Medical University, Beijing 100026, China; 2Translational Medicine Research Center, Medical Innovation Research Division of Chinese PLA General Hospital, Beijing 100039, China; 3Prenatal Diagnosis Center, Shijiazhuang Obstetrics and Gynecology Hospital, Shijiazhuang 050011, China; 4Jiaen Genetics Laboratory, Beijing Jiaen Hospital, Beijing 100083, China; 5Department of Ultrasound, Beijing Obstetrics and Gynecology Hospital, Capital Medical University, Beijing 100026, China

**Keywords:** Osteogenesis imperfecta, *COL1A1*, *COL1A2*, *CANT1*, *RMRP*

## Abstract

Introduction: Osteogenesis imperfecta (OI) is a rare mendelian skeletal dysplasia with autosomal dominant or recessive inheritance pattern, and almost the most common primary osteoporosis in prenatal settings. The diversity of clinical presentation and genetic etiology in prenatal OI cases presents a challenge to counseling yet has seldom been discussed in previous studies. Methods: Ten cases with suspected fetal OI were enrolled and submitted to a genetic detection using conventional karyotyping, chromosomal microarray analysis (CMA), and whole-exome sequencing (WES). Sanger sequencing was used as the validation method for potential diagnostic variants. In silico analysis of specific missense variants was also performed. Results: The karyotyping and CMA results of these cases were normal, while WES identified OI-associated variants in the *COL1A1/2* genes in all ten cases. Six of these variants were novel. Additionally, four cases here exhibited distinctive clinical and/or genetic characteristics, including the situations of intrafamilial phenotypic variability, parental mosaicism, and “dual nosogenesis” (mutations in collagen I and another gene). Conclusion: Our study not only expands the spectrum of *COL1A1/2*-related OI, but also highlights the complexity that occurs in prenatal OI and the importance of clarifying its pathogenic mechanisms.

## 1. Introduction

Osteogenesis imperfecta (OI), also known as “brittle bonedisease”, is a spectrum of rare inherited connective-tissue disorders mainly characterized by fractures with minimal or absent trauma, variable dentinogenesis imperfecta (DI), and, in adult years, hearing loss [1]. The incidence of OI is approximately 6–7 per 100,000 births [2]. OI is phenotypically and genetically heterogeneous with clinical severity ranges from perinatal lethal to nearly asymptomatic [2]. Based on clinical presentations, radiographic features, family history, and natural history, OI was initially classified into four common types by Sillence et al. [3], and has recently been re-categorized into 18 fine subtypes according to its genetic pathogenesis [4].

To date, at least 20 genes with autosomal dominant or recessive inheritance pattern have been recognized to be responsible for OI, including *COL1A1*, *COL1A2*, *BMP1*, *SERPINF1*, *SERPINH1*, *CRTAP*, *P3H1*, *PPIB*, *TMEM38B*, *WNT1*, *FKBP10*, *PLOD2*, *IFITM5*, *MBTPS2*, *CREB3L1*, *SP7*, *SPARC*, *P4HB*, *PLS3*, and *SEC24D* [1], of which the *COL1A1/2* (MIM *120150/120160) associated subtypes account for ~90% cases [2]. As the main etiological components, *COL1A1/2* encode the alpha 1 and alpha 2 chains of type I collagen, the major protein component of the extracellular matrix inbone, skin, and tendon. Untilnow, over 3000 variants in *COL1A1/2* have been indexed in the osteogenesis imperfecta variant database, yet the phenotype–genotype correlation in this disorder is still not meticulously established (http://www.le.ac.uk/ge/collagen/; accessed on 25 September 2020).

In the clinical practice of prenatal diagnosis, ultrasonographic evaluation is the first-line screening method for skeletal dysplasias (SD) during both early and late gestational stages [5]. Yet, it has obvious shortcomings in the defining of disease types and possible prognosis, which requires the facilitation of genetic techniques. In our previous study, we established a trio whole-exome sequencing (trio-WES, meaning to submit “proband–father–mother” trios simultaneously for sequencing and mutation screening analysis) detection strategy, introduced it into the diagnosis of prenatal SD cases, and achieved a considerable detection efficiency [6,7]. We also reported a rare autosomal recessive OI case and identified a novel compound heterozygous variation in the *PLOD2* gene, which encodes the lysine hydroxylase responsible for the proper cross-linking of type I collagen pro-fibrils [8]. Continuous effort has been made to better understand the various conditions of prenatal OI.

Here in the present study, 10 cases with prenatal skeletal dysplasia were enrolled and underwent a comprehensive clinical and genetic investigation. All the patients were positive with *COL1A1* or *COL1A2* variants; however, distinct situations existed in different cases, including the intrafamilial phenotypic variability, parental mosaicism, and “dual nosogenesis” with variants in other genes. Our study highlights the complexity of *COL1A1*/*COL1A2* associated OI, especially in the prenatal cases.

## 2. Materials and Methods

The present study was approved by the Ethics Committee of Beijing Obstetrics and Gynecology Hospital (No. 2018-KY-003-01). Informed consent wasprovided by all the participants.Allprocedures performed in the present studywere in accordance withtheDeclaration of Helsinki 1964 and its later amendments orcomparable ethical standards.

### 2.1. Subjects and Clinical Evaluation

Between December 2018 and January 2021, a total of 10 cases with fetuses displaying *in utero* skeletal dysplasia were enrolled in our centers. Clinical evaluation was made via the combination of ultrasound monitoring at mid-trimester X-ray examination after fetus’ abortion, and a comprehensive survey on family history.

### 2.2. Karyotyping and Copy Number Variation (CNV) Analysis

To conduct a prenatal genetic diagnosis, amniocentesis was performed to obtain the amniotic fluid (AF) samples according to the routineprocess. Conventional chromosomal karyotyping by G-binding was performed to detect overall chromosomal anomalies [9].

Genomic DNA was extracted from fetal samples and their parents’ peripheral blood using QIAamp DNA Midi Kit (Qiagen, Dusseldorf, Germany). Chromosomal microarray analysis (CMA) tests with CytoScan 750K SNP-array (Affymetrix Inc., Santa Clara, CA, USA) were carried out accordingto the manufacturer’s manual workflow on all fetal specimensin order to investigate genomic CNVs with clinical significance.

### 2.3. Whole-Exome Sequencing

Whole-exome sequencing (WES) was employed to detect thesequence variants inthe probands’ fetal samples, as describedin our previous study [10]. Briefly, the target-region sequence enrichment was performed using the Agilent Sure Select Human Exon Sequence Capture Kit (Agilent, Palo Alto, CA, USA).DNA libraries were tested by quantitative PCR, where the size, distribution and concentration were determined using Agilent Bioanalyzer 2100 (Agilent, Palo Alto, CA, USA). Along with ~150 bp pair-endreads, the NovaSeq-6000 platform (Illumina, Inc., San Diego, CA, USA), was used for sequencing of DNA with~300 pM per sample with NovaSeq Reagent kit. Sequencing raw reads (quality level Q30%>90%; and the quality criteria was listed at https://www.illumina.com/science/technology/next-generation-sequencing/plan-experiments/quality-scores.html; accessed on 25 September 2020) were aligned to the human reference genome (accession No. hg19/GRCh37) using the Burrows Wheeler Aligner tool, and PCR duplicates were removed using Picardv1.57. Variant calling was performed with the Verita Trekker^®^ Variants Detection system (v2.0; Berry Genomics, Beijing, China) and Genome Analysis Tool Kit (https://software.broadinstitute.org/gatk/; accessed on 25 September 2020). Then, variants were annotated and interpreted using ANNOVAR (v2.0) [11] and Enliven^®^ Variants Annotation Interpretation systems (Berry Genomics), based on the common guidelines by ACMG (American College of Medical Genetics and Genomics) [12].To assist withthe interpretation of variant pathogenicity, we referred to three frequency databases (ExAC_EAS, http://exac.broadinstitute.org, accessed on 25 September 2020; gnomAD_exome_EAS, http://gnomad.broadinstitute.org, accessed on 25 September 2020; 1000G_2015aug_eas, https://www.internationalgenome.org, accessed on 25 September 2020) and HGMD (Human Gene Mutation Database) pro v2019; Revel score (a combined method of pathogenicity prediction) [13] and pLI score (representing the tolerance for truncating variants) were also employed.

Sanger sequencing was performed as a validation method with the 3500DX Genetic Analyzer (Applied Biosystems, Thermofisher, LA, CA, USA).

### 2.4. Conservatism and Structure Analysis

The evolutionary conservatism of all affected amino acid (AA) residues by corresponding missense variants was analyzed using the online tool, MEGA7 (http://www.megasoftware.net/previousVersions.php; accessed on 25 September 2020), with default parameters. Structural analysis and molecular dynamics simulation of a novel missense variant detected in the *CANT1* gene were presented in Appendix A.

## 3. Results

### 3.1. Clinical Manifestations

In the 10 recruited families, the average maternal age was 31.2 (ranging from 26 to 38), and the gestational weeks with initial SD diagnosis ranged from 16W to 23W4D. All the couples were non-consanguineous and claimed to have no family history of genetic disorders. Most fetuses showed OI-like characteristics in the second or third gestational trimester, e.g., the bending or angulation of the limb long bones. However, there were a few cases with unique phenotypes, such as Case 7 and Case 8. In Case 7, the fetus presented with tibiofibula and ulnar flexure dysplasia, poor skull ossification, knee bendingabnormity, bilateral short choroid plexus; in Case 8, the fetus additionally showed thickened NT (nuchal translucency = 0.6 cm), anasarca, abnormal ankle joint and foot posture at the early second trimester. Empirically, these symptoms are highly unusual in common OI cases. According to our pregnancy tracking, all the fetus probands were aborted.

The detailed clinical manifestations and information of the 10 cases were listed in Table 1. The representative clinical images of affected fetuses were demonstrated in Figure 1, and the diagram of each case pedigree was shown in Figure 2 and Figure 3.

### 3.2. Genetic Findings

All 10 cases were negative for G-banding karyotyping and CMA detection. Subsequently, WES identified 10 specific diagnostic variants in these cases, which were distributed in the *COL1A1* and *COL1A2* genes, namely *COL1A1*: c.1678G>A (p.Gly560Ser) in Case 1, *COL1A1*: c.2101G>A (p.Gly701Ser) in Case 2, *COL1A1*: c.3557C>T (p.Pro1186Leu) in Case 3, *COL1A2*: c.856G>C (p.Gly286Arg) in Case 4, *COL1A2*: c.1072G>A (p.Gly358Ser) in Case 5, *COL1A2*: c.2198G>T (p.Gly733Val) in Case 6, *COL1A2*: c.2295+1(IVS37)G>C in Case 7, *COL1A1*: c.2300G>T (p.Gly767Val) in Case 8, *COL1A2*: c.2783G>A (p.Gly928Asp) in Case 9, and *COL1A2*: c.3052G>A (p.Gly1018Ser) in Case 10. Table 2 included the details of all these *COL1A1/2* variants. Sanger sequencing verified these variants and their familial carrying status, and the results were demonstrated in Figure 2 and Figure 3. Except for the variant in Case 3, all the variants were *de novo*. Six novel variants amongst them were reported, which were *COL1A1*: c.3557C>T, *COL1A2*: c.856G>C, *COL1A2*: c.2295+1(IVS37)G>C, *COL1A1*: c.2300G>T, *COL1A2*: c.2783G>A, and *COL1A2*: c.3052G>A. 

There were four cases with special circumstances that required special attention. Case 3: Three subjects (the father and two fetuses) harboring the identical *COL1A1*: c.3557C>T variant showed a strong variable expressivity. The father was asymptomatic, while both fetuses had typical OI characteristics. Case 7: An additional heterozygous compound variation in the *CANT1* gene, consisting of c.556G>A and c.-220G>A variants inherited from the parents, respectively, was identified and considered to be with contribution to the fetus’ phenotype (Figure 3, Family 7; Table 2). Case 8: A heterozygous compound variation in the *RMRP* gene, consisting of n.39A>G and n.23C>T variants inherited from the parents, respectively, was identified and considered to contribute to the fetus’ phenotype (Figure 3, Family 8; Table 2). Case 9: The mother, who was asymptomatic, was a mosaic carrier of the *COL1A2*: c.2783G>A variant (Figure 3, Family 9).

### 3.3. Analysis of the Missense Variants

In this study, a total of nine missense variants in *COL1A1/2* genes were detected. In addition, according to the analysis results, the AA residues they affected maintained a high degree of evolutionary conservatism among species (Figure 4). Since the pathogenicity of the CANT1: Val186Ile variant was not certain, we conducted the structural and molecular dynamics analysis. It was demonstrated that this variant probably impacted the protein stability and secondary structure (detailed results see Appendix A).

## 4. Discussion

Osteogenesis imperfecta (OI) is a rare genetic disorder with an estimated prevalence between 1/13,500 and 1/9700(14), yet the most common form of primary osteoporosis in prenatal and perinatal circumstances [7,14,15]. There were numerous studies using molecular genetic techniques to achieve the prenatal diagnosis on OI [8,16,17,18,19]. A great deal of mutations in the *COL1A1/2* genes, which confirms to the autosomal dominant pattern and contributed to a major part of OI cases, have been reported and indexed; as a result, researchers have begun to develop methods to assess the prognosis caused by specific types of mutations in recent years [20]. However, in the prenatal settings, OI cases may present with more complexity, which still warrants more sufficient attention. In the present study, we retrospectively summarized the clinical and genetic characteristics of 10 cases with *COL1A1/2*-related OI. Generally speaking, the affected fetuses were diagnosed by ultrasonography at middle to late trimester with characteristic long bone curvature. Among all the *COL1A1/2* variations, missense variants resulting in the replacement of Gly residue in the triple helical domain occupied the major part, up to eight in total, which indicates the predominant role of these variations in OI pathogenesis. Besides, the conservatism of AA residues affected by each missense variant strongly supports their pathogenicity. On the other hand, some individual cases displayed unique clinical presentations, which could likely be related to their specific genetic backgrounds.

In Case 3, the asymptomatic father and two affected fetuses all carried the missense variant COL1A1: p.Pro1186Leu, which reflects the strong intra-familial phenotypic variability of OI. According to the large cohort studied by Zhytniket al., 32.81% families had increasing or decreasing OI phenotype severity across generations, and higher intrafamilial variability of phenotypes correlated with the collagen I dominant negative variants [21]. In other words, the father in the family may have cryptic mild skeletal symptoms that went undetected by the clinician. As for this novel variant, the pathogenicity of another missense variation (p.Pro1186Ala) affecting the identical residue has been reported [22], thus supporting it to be deleterious. It was reported that dominant-negative mutations in *COL1A1* were more likely to result in phenotypic variability, but functional experiments are needed to test this theory and provide evidence of whether specific variations are more suitable for particular treatment [21,23].

In Case 7, the affected fetus presented with a very complex and extensive skeletal dysplasia, while a suspected causative compound heterozygous variations in the *CANT1* gene was detected, along with the splicing site variant in *COL1A2*. The *CANT1* gene (Calcium-activated nucleotidase 1, MIM *613165), encoding an extracellular protein that functions as a nucleotide tri- and diphosphatase [24], is responsible for a spectrum of skeletal dysplasia with diverse indications, including the Desbuquois dysplasia 1 (MIM #251450) [25], the multiple epiphyseal dysplasia 7 (MED, MIM #617719) [26], and the pseudodiastrophic dysplasia (PDD, MIM #264180) [27]. Among them, PDD is the rarest subtype and associated with prenatal manifestation and early lethality, characterized by short-limbed short stature, facial dysmorphism, and distinctive skeletal abnormalities including short ribs, mild tomoderate platyspondyly, broad ilia with flaring, increased acetabular angle, shortened long boneswith metaphyseal flaring, elongation of the proximal and middle phalanges with subluxation of theproximal interphalangeal joints, subluxation of the elbow, and talipes equinovarus [28,29]. These descriptions from previous reports fit well with the fetal phenotype in Case7, so it was supposed that this variation could also most likely contribute to its manifestation. In that case, fertility counseling for the couple would face greater challenges. Ideally, the couple would need to find a way to avoid the c.2295+1(IVS37)G>C variant and the co-existing of both variants in *CANT1* in their future pregnancies, which would be difficult to achieve.

In Case 8, the situation was similar. Besides the Glycine substitution in *COL1A1*, another heterozygous compound variation in the *RMRP* gene was identified. *RMRP* (RNA component of mitochondrial RNA processing endoribonuclease, MIM *157660) can be transcribed into a long non-coding RNA, which is associated with a wide spectrum of autosomal recessive skeletal conditions, ranging from the mild metaphyseal dysplasia without hypotrichosis (MDWH, MIM #250460) [30] and cartilage-hairhypoplasia (CHH, MIM #250250) [31] to the severe anauxetic dysplasia 1 (ANXD1, MIM #607095) [32]. This clinical spectrum includes different degrees of short stature, hair hypoplasia, defective erythrogenesis, and immunodeficiency, and the mutant RMRP may affect both messenger RNA (mRNA) and ribosomal RNA (rRNA) cleavage and thus cell-cycle regulation and protein synthesis [33]. In terms of morphology, the fetus in Case 8 had similar characteristics to ANXD1, such as short stature and multiple skeletal dysplasia; meanwhile, it also exhibited unique prenatal indications at early pregnancy, including NT thickening and systemic edema. Admittedly, the two variants of *RMRP* are novel and need to be validated by functional experiments; however, we do support the idea that variations in both *COL1A1* and *RMRP* contribute to the fetal phenotype. In a large cohort study, Posey et al. pointed out that cases involving two-gene pathogenic variants were not uncommon, and that the most typical case is a compound heterozygous variationin a recessive gene plus a *de novo* variation in a dominant gene [34]. We speculate that Case7 and Case8 should fit this scenario. Therefore, with regard to their future pregnancies, special care should be taken, and interventional assisted reproduction may be best recommended.

In Case 9, a novel missense variant leading to the Glycine substitution in *COL1A2* was detected, and Sanger sequencing revealed that this variant was inherited from the asymptomatic mosaic mother. Parental mosaicism, whether confined to the genital system or not, has been reported in OI from time to time and should be noted in reproductive counseling [35,36,37]. According to our follow-up, the couple chose preimplantation diagnosis and selected a wild-type embryo. At present, the pregnancy has reached 38 weeks and the prenatal phenotype is normal.

The main limitation of this study is the lack of adequate functional experimental verification for several specific mutations with uncertain clinical significance at genetic level. In conclusion, this pilot study presents a landscape of complicated prenatal *COL1A1/2*-OI cases. We identified six novel variants in these two collagen genes, as well as compound variations in two other genes associated with skeletal dysplasia. However, most importantly, our findings suggest the need to be particularly aware of these specific conditions during prenatal and reproductive counseling in OI case.

## Figures and Tables

**Figure 1 genes-13-01578-f001:**
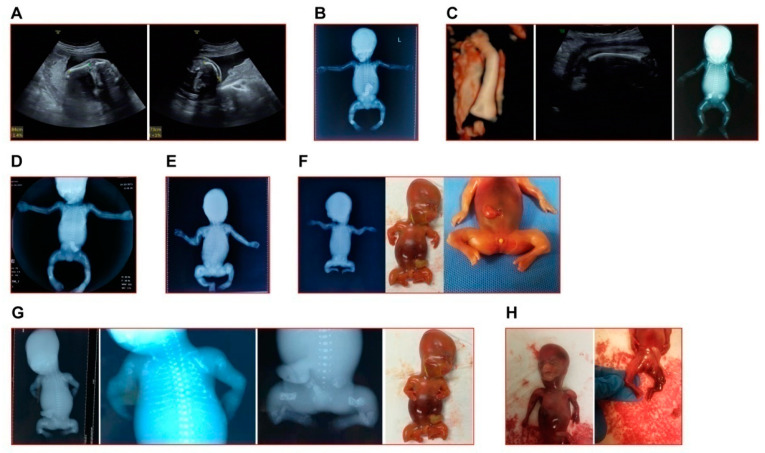
The representative clinical images of fetuses in this study. (**A**) The left humerus (left image) and right femur (right image) of fetus in Case 1 appeared curved at angled. (**B**) The X-ray image of induced fetus in Case 2 showed typical OI indications (short and curved femur, tibia and fibula). (**C**) Case 3: left and middle, the 3D reconstruction of the curved femur and the in utero curve of femur of the second fetus; right, the X-ray image of the first induced fetus in this family. (**D**) The X-ray image of induced fetus in Case 5. (**E**) The X-ray image of induced fetus in Case 6. (**F**) The X-ray image (left) and appearance (middle and right) of induced fetus in Case 7. (**G**) The X-ray images and appearance (farthest to the right) of induced fetus in Case 8. (**H**) The appearance of induced fetus in Case 9. **Note:** In accordance with the wishes of the affected families, clinical images of Cases 4 and 10 are not shown.

**Figure 2 genes-13-01578-f002:**
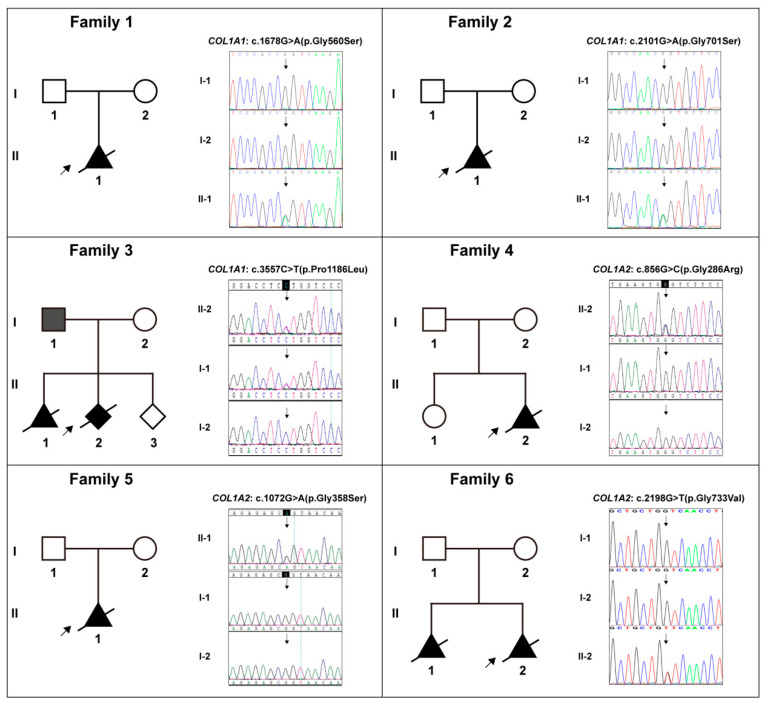
The pedigree diagrams of Cases 1–6 (Families 1–6) and the variants in collagen genes (*COL1A1/2*) in Sanger sequencing form and their corresponding carrying status in each family. **Note:** Solid shapes represent mutation carriers; the arrow points to the probands; the diagonal line means dead.

**Figure 3 genes-13-01578-f003:**
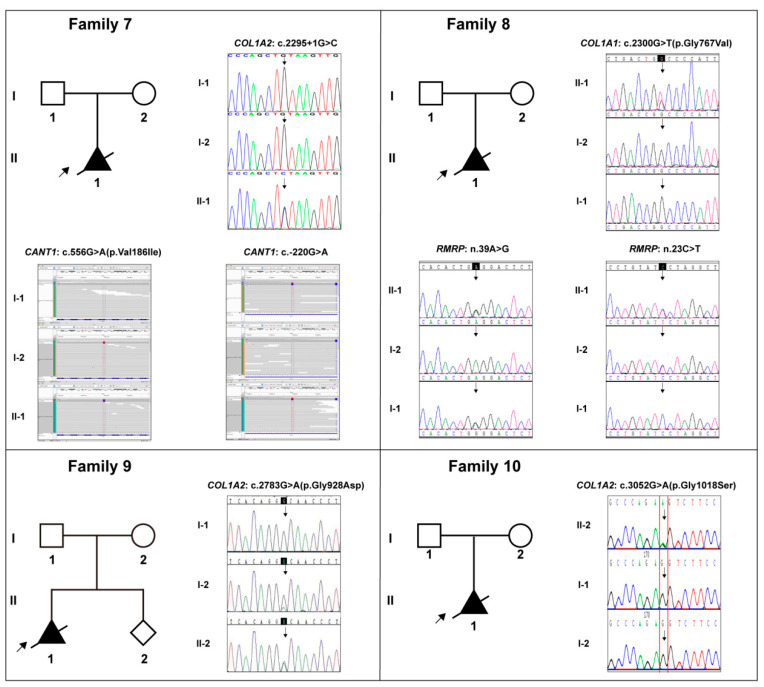
The pedigree diagrams of Cases 7–10 (Families 7–10) and the variants in collagen genes (*COL1A1/2*), *CANT1* gene, and *RMRP* gene in Sanger sequencing or Bam form, and their corresponding carrying status in each family. **Note:** Solid shapes represent mutation carriers; the arrow points to the probands; the diagonal line means dead.

**Figure 4 genes-13-01578-f004:**
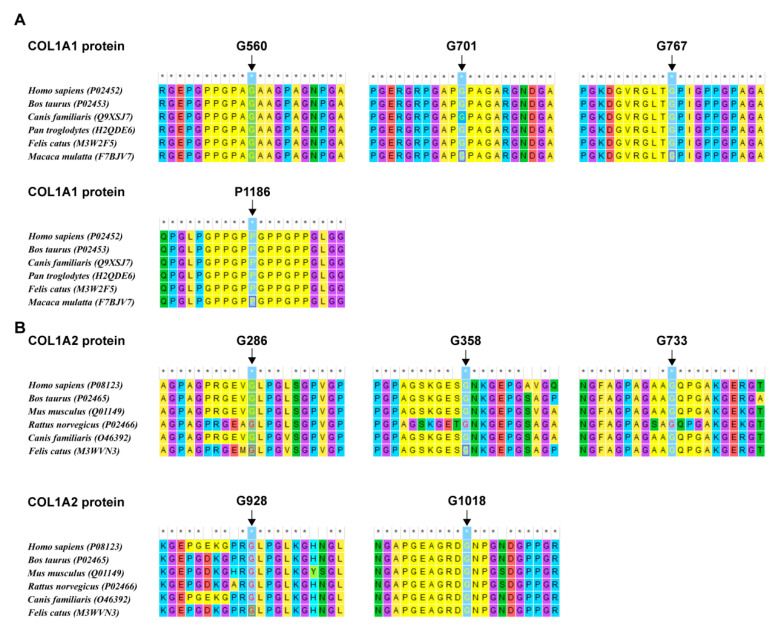
The evolutionaryconservatism of AA residues affected by nine missense variants in the *COL1A1/2* genes identified in this study. (**A**) Conservatism of the residues in COL1A1 affected by missense variants detected in this study. (**B**) Conservatism of the residues in COL1A2 affected by missense variants detected in this study.

**Table 1 genes-13-01578-t001:** Clinical information of the ten recruited cases.

Case No.	Maternal Age (Years)	Gestational Age with Initial Diagnosis *	Clinical History *	Fetal Sample for WES
1	28	23W4D	G1P0; Ultrasound examination revealed that the long bones of the fetus’s limbs were short and curved; Fetus aborted at 24W1D.	Umbilical cord
2	35	22W6D	G1P0; Fetal femur, tibia and fibula were initially found to be short and curved; Fetus aborted at 23W5D.	Umbilical cord
3	26; 27 (two pregnancies)	22W; 20W6D (two pregnancies)	G3P0; Two affected pregnancies: (1) The right femur was initially found to be “telephone like” at 22W; Both femurs were identified as short and curved at 24W; Fetus aborted at 25W; (2) Both femurs were identified as short and curved at 20W6D; Fetus aborted at 32W; (3) Still pregnant before submission.	Amniotic fluid; Umbilical cord
4	36	20W1D	G2P1 (A normal daughter at 5years old); Limb long bones short and curved, some of the ribs recessed inward at 20W1D; Fetus aborted at 23W.	Umbilical cord
5	26	20W2D	G1P0; Limb long bones short and curved at 20W2D; Fetus aborted at 25W; Autopsy revealed blue staining of the sclera and a thin, soft skull.	Umbilical cord
6	38	20W5D	G2P0 (One miscarriage 5 years ago); The fetus was found to have short and curved limb long bones and left foot varus at 20W5D; Fetus aborted at 23W3D.	Umbilical cord
7	31	16W	G1P0; The fetus was found presenting with short limb long bones, dysplasia of tibiofibula and ulnar flexure, poor ossification of the skull, abnormal knee bending, bilateral short choroid plexusat 16W; Fetus aborted at 17W1D.	Umbilical cord; Skin tissue
8	35	13W4D	G1P0; NT (nuchal translucency) thickening (6.0 mm), anasarca, limb long bones short and curved, abnormal ankle joint and foot posture at 13W4D; Fetus aborted at 17W.	Amniotic fluid; Umbilical cord
9	30	16W1D	G2P0; Limb long bones short at 16W1D; Definitive ultrasound diagnosis was made at 22 weeks; Fetus aborted at 23W. The next pregnancy was via pre-implantation diagnosis and is now with normal phenotype at 30W.	Amniotic fluid
10	29	20W4D	G1P0; Limb long bones short and curved at 20W4D; Fetus aborted at 23W.	Umbilical cord

* W: weeks; D: days; G: gravida; P: para; A: abortus.

**Table 2 genes-13-01578-t002:** Information of the genetic variations identified in this study.

Case No.	Gene *	DNA Variation	Protein Variation	HGMD * Rating (PMID)	Frequency in Three Databases *	Revel Prediction *	Pathogenicity Level * (Evidences)
1	*COL1A1*	c.1678G>A	p.Gly560Ser	DM (15741671)	-; -; -	0.987	P (pp2+pm2+pm5_strong+ ps4_supporting+ps2+pp3)
2	*COL1A1*	c.2101G>A	p.Gly701Ser	DM (17078022)	-; -; -	0.991	P (pp2+pm2+pm5_strong +ps2+pp3)
3	*COL1A1*	c.3557C>T	p.Pro1186Leu	/	-; -; -	0.555	VUS (pp2+pm2+pm5)
4	*COL1A2*	c.856G>C	p.Gly286Arg	/	-; -; -	0.988	LP(pm2+pm5_strong+pp3)
5	*COL1A2*	c.1072G>A	p.Gly358Ser	DM (9240878)	-; -; -	0.981	LP(pm2+pm5_strong+pp3)
6	*COL1A2*	c.2198G>T	p.Gly733Val	DM (25086671)	-; -; -	0.993	VUS (pm2+pm5+pp3)
7	*COL1A2*	c.2295+1G>C		/	-; -; -	/	LP(pvs1+pm2)
	*CANT1*	c.556G>A	p.Val186Ile	/	0.001;0.002082;0.0029362	0.272	VUS (pm2)
	*CANT1*	c.-220G>A	/	/	-; -; -	/	VUS
8	*COL1A1*	c.2300G>T	p.Gly767Val	/	-; -; -	1.000	LP(pp2+pm2+pm5+pp3)
	*RMRP*	n.39A>G	/	/	-; -; -	/	VUS
	*RMRP*	n.23C>T	/	/	-; -; -	/	VUS
9	*COL1A2*	c.2783G>A	p.Gly928Asp	/	-; -; -	0.983	LP (pm2+pp3)
10	*COL1A2*	c.3052G>A	p.Gly1018Ser	/	-; -; -	0.960	VUS (pm2+pp3+pm6)

* Gene transcripts No.: *COL1A1*, NM_000088.3; *COL1A2*, NM_000089.3; *CANT1*, NM_001159773; *RMRP*, NR_003051.3; HGMD: Human Gene Mutation Database (Professional Version 2021.10); PMID: PubMed ID (https://pubmed.ncbi.nlm.nih.gov; accessed on 27 September 2020); Three databases: 1000g2015aug_eas (https://www.internationalgenome.org; accessed on 25 September 2020);ExAC_EAS (http://exac.broadinstitute.org; accessed on 25 September 2020); gnomAD_exome_EAS (http://gnomad.broadinstitute.org; accessed on 25 September 2020); Revel: An ensemble method for predicting the pathogenicity of missense variants on the basis of individual tools: MutPred, FATHMM, VEST, PolyPhen, SIFT, PROVEAN, MutationAssessor, MutationTaster, LRT, GERP, SiPhy, phyloP, and phastCons (http://dx.doi.org/10.1016/j.ajhg.2016.08.016); Pathogenicity level rating: By ACMG (The American College of Medical Genetics and Genomics); P: pathogenic; LP: likely pathogenic; VUS: variants of unknown significance.

## Data Availability

All the data involved in this study were provided in the manuscript, tables and figures.

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
