# Peer review of "Prenatal Cases Reflect the Complexity of the COL1A1/2 Associated Osteogenesis Imperfecta"

_genes, 2022, doi:10.3390/genes13091578_

Round 1

Reviewer 1 Report

The manuscript entitled “Prenatal Cases Reflect the Complexity of the COL1A1/2 Associated Osteogenesis Imperfecta“ investigates the effect of COL1A1/COL1A2 variants associated OI in 10 cases using a combination of clinical evaluations, CMA, WES and Sanger sequencing followed by conservatism and structure analysis of all affected amino acid residues.

To conduct the prenatal genetic analysis, the authors mentioned that the amniotic fluid was used (Materials and Methods, line 83), however in table 1, the show that the fetal samples were harvested from other tissues, mostly from the umbilical cord. Please explain how the sample’s origin may affect the results and why the protocol was not standardized for all fetuses.

Please refer to figure 1 in 3.1, first paragraph.

Figure 1 is missing representative images of case 4 and 10.

Reviewer 2 Report

The authors Yang et al. reported the complexity of COL1A1/2 in prenatal osteogenesis imperfecta. The manuscript is well-written with only minor mistakes can be found. Here are some comments to improve the overall quality of the manuscript.

1.       Abstract: The abbreviation “WES” should be defined.

2.       Materials and Methods: The ethical approval code should be included.

3.       Results: Clarification/description is need for case #3, whereby there are two maternal age and gestation age with initial diagnosis.

4.       Figure 2 and 3: Why authors separated family 1-6 and 7-10 into two different figures, although the results are obtained using similar approach? In certain cases, CANT1 and RMRP genes can be detected but not for all families. Why?

5.       It is recommended to provide appropriate indication in the pedigree to ensure better understanding by non-expert.

Reviewer 3 Report

The authors provide a well-structured manuscript, where they report relevant findings in 10 cases of prenatal osteogenesis imperfecta. In all cases they identified variants in either COL1A1 or COL1A2, yet two of them additionally carried variants in other skeletal disease genes. Although the manuscript is primarily descriptive, the clarity of data presentation is excellent, and there is indeed a true necessity to increase our understanding of prenatal skeletal disorders. In my opinion, there is only one issue that remains to be clarified, i.e. the lack of Supplementary Material, as this was stated in line 203.
